# Offseason Body Composition Changes Detected by Dual-Energy X-ray Absorptiometry versus Multifrequency Bioelectrical Impedance Analysis in Collegiate American Football Athletes

**DOI:** 10.3390/sports9080112

**Published:** 2021-08-19

**Authors:** Jake R. Boykin, Grant M. Tinsley, Christine M. Harrison, Jessica Prather, Javier Zaragoza, Matthias Tinnin, Shay Smith, Camden Wilson, Lem W. Taylor

**Affiliations:** 1Energy Balance & Body Composition Laboratory, Department of Kinesiology & Sport Management, Texas Tech University, Lubbock, TX 79409, USA; jake.boykin@ttu.edu (J.R.B.); grant.tinsley@ttu.edu (G.M.T.); 2Human Performance Laboratory, School of Exercise & Sport Science, University of Mary Hardin-Baylor, Belton, TX 76513, USA; cmharrison@mail.umhb.edu (C.M.H.); jprather@mail.umhb.edu (J.P.); mtinnin@umhb.edu (M.T.); smsmith3@mail.umhb.edu (S.S.); ccwilson@mail.umhb.edu (C.W.); 3School of Kinesiology, Applied Health and Recreation, Oklahoma State University, Stillwater, OK 74078, USA; jazarag@okstate.edu

**Keywords:** fat-free mass, body fat, fat-free mass index, body fat percentage, DXA, BIA

## Abstract

Tracking changes in body composition may provide key information about the effectiveness of training programs for athletes. This study reports on the agreement between bioelectrical impedance analysis (BIA) and dual-energy X-ray absorptiometry (DXA) for tracking body composition changes during a seven-week offseason training program in 29 NCAA collegiate American football players. Body composition in subjects (mean ± SD; age: 19.7 ± 1.5 y; height: 179.8 ± 6.6 cm; body mass (BM: 96.1 ± 12.6 kg; DXA body fat: 20.9 ± 4.4%) was estimated using BIA (InBody 770) and DXA (Hologic Horizon) before and after the training intervention. Repeated measures ANOVA and post hoc comparisons were performed. Longitudinal agreement between methods was also examined by concordance correlation coefficient (CCC) and Bland–Altman analysis alongside linear regression to identify bias. Significant method by time interactions were observed for BM (DXA: 1.1 ± 2.4 kg; BIA: 1.4 ± 2.5 kg; *p* < 0.03), arms fat-free mass (FFM) (DXA: 0.4 ± 0.5 kg; BIA: 0.2 ± 0.4 kg; *p* < 0.03), and legs FFM (DXA: 0.6 ± 1.1 kg; BIA: 0.1 ± 0.6 kg; *p* < 0.01). Post hoc comparisons indicated that DXA—but not BIA—detected increases in FFM of the arms and legs. Time main effects, but no method by time interactions, were observed for total FFM (DXA: 1.6 ± 1.9 kg; BIA: 1.2 ± 2.1 kg; *p* = 0.004) and trunk FFM (DXA: 0.7 ± 1.3 kg; BIA: 0.5 ± 1.0 kg; *p* = 0.02). Changes in total BM (CCC = 0.96), FFM (CCC = 0.49), and fat mass (CCC = 0.50) were significantly correlated between BIA and DXA. DXA and BIA may similarly track increases in whole-body FFM in American collegiate football players; however, BIA may possess less sensitivity in detecting segmental FFM increases, particularly in the appendages.

## 1. Introduction

In collegiate American football programs, strength and conditioning practitioners often design the initial period of offseason training to prioritize increases in fat-free mass (FFM) and muscular performance in athletes. This period allows for recovery of FFM, as research suggests that athletes experience significant reductions in FFM over the course of a competitive season [1,2]. Such an emphasis on increasing FFM is also prudent according to research that suggests fat-free mass index (FFMI) may be related to football performance [3,4]. Although the physical demand of the sport varies widely by position, players at every position may benefit from increasing FFM [5]. Improvements in football performance due to favorable FFM changes are likely related to their beneficial effects on relevant performance outcomes, such as strength and power [6,7]. These variables are particularly important in college football and can be used to differentiate players by division [4,8] and predict starters and non-starters [9]. These exercise performance outcomes are also assumed to be predictive of sport performance at the professional level, contributing to an increased focus on bench press and jump performance at the National Football League (NFL) combine.

Due to continued interest in the relationship between body composition and football performance, there is an emergent incentive for strength and conditioning coaches to monitor body composition in their athletes in response to training programs. While there are several technologies that are able to estimate body composition, dual-energy X-ray absorptiometry (DXA) is a well-accepted laboratory method that is often used as a reference for estimating total and segmental body composition. However, despite its popularity, DXA has some disadvantages that limit its implementation in some settings. These include the high cost, minor radiation exposure, and lack of portability. Due to the unavailability of DXA in some settings, an examination of alternate methods is warranted. In this regard, bioelectrical impedance analysis (BIA) has garnered widespread interest and adoption due to its relatively low cost and ease of use [10]. To date, conflicting results have been observed when comparing DXA and BIA body composition changes [11,12]. Most studies have observed disagreement between longitudinal body composition estimates among methods, although these have focused on periods of weight loss in obese populations [13,14,15]. The comparability of BIA and DXA for quantifying body composition changes during periods of purposeful weight gain in athletes is not fully established.

Recently, two studies compared DXA and BIA body composition changes in subjects undergoing a resistance training program and found that estimations of total body composition changes were similar between BIA and DXA [16,17]. Interestingly, one study comparing DXA and BIA to a rapid four-compartment model indicated that BIA demonstrated lower errors than DXA for tracking body composition changes when subjects increased both FFM and fat mass (FM) during a resistance training program [17]. While there have been recent studies examining longitudinal changes in body composition in football athletes, these studies have relied on DXA alone [1,18]. Recently, another comparison between BIA and DXA in elite rugby players showed that BIA and DXA tracked similar changes in FFM and FM [19]. If changes in body composition can be accurately assessed by BIA during a period of resistance training in football athletes, then the capacity and accessibility of estimating changes in body composition in athletic settings would be meaningfully increased. Therefore, the goal of the present study was to quantify the relationship between body composition changes estimated by DXA and BIA in American football players during the initial period of an offseason training program, for the purpose of determining how comparable changes are when estimated by these distinct methods. It was hypothesized that total body composition changes detected by DXA and BIA would not significantly differ but that discrepancies in segmental changes may be observed.

## 2. Materials and Methods

This study assessed the agreement of body composition estimates obtained by BIA and DXA in NCAA Division III American football players. Total and segmental body compositions were measured by BIA and DXA in 29 football athletes before and after a seven-week early offseason training program. Body composition estimates were compared between methods to determine the comparability of the methods for tracking football-training-mediated changes in FFM and FM.

### 2.1. Subjects

Male Division III football players were recruited for participation in this study. Prior to testing, subjects were pre-screened to rule out any exclusion criteria. Subjects were excluded if they were not a current member of the football team, if they experienced any adverse events during the study and/or if they were not cleared to participate in the offseason training. Following the completion of a medical and health history questionnaire, subjects were familiarized with testing protocols and signed an informed consent statement approved by the University of Mary Hardin-Baylor’s Institutional Review Board. Total sample size was limited by the number of consenting athletes in the football program during the single offseason period. Twenty-nine subjects (age: 19.8 ± 1.5 y; height: 179.8 ± 6.6 cm; baseline scale BM: 96.1 ± 12.6 kg; baseline DXA BF%: 20.9 ± 4.4%; baseline BIA BF%: 20.5 ± 5.8%; baseline DXA fat-free mass index (FFMI): 23.1 ± 2.1 kg/m^2^) were included in the body composition analysis. Players were divided into two position-specific groups: BIGS (offensive linemen, defensive linemen, tight ends, and linebackers) and SKILLS (all remaining positions). The baseline characteristics of BIGS (*n* = 22) were: height = 180.6 ± 6.5 cm; scale BM = 99.8 ± 11.4 kg; DXA BF% = 22.5 ± 3.9%; BIA BF% = 22.3 ± 5.4%; DXA FFMI = 23.4 ± 2.3 kg/m^2^). The baseline characteristics of SKILLS (*n* = 7) were: height = 177.4 ± 7.0 cm; scale BM = 84.7 ± 9.2 kg; DXA BF% = 16.1 ± 1.6%; BIA BF% = 14.9 ± 2.5%; DXA FFMI = 22.2 ± 1.1 kg/m^2^). The number of subjects completing performance testing ranged from 16 to 19 per assessment due to injuries or other contraindications for testing.

### 2.2. Procedures

Testing timepoints took place during the initial period of the Division III football offseason (January to March). Initial measurements (Pre) served to provide baseline body composition values and subsequent measurements (Post) provided body composition data following the implementation of a seven-week resistance training mesocycle. To promote consistent conditions across all timepoints, athletes were instructed to fast for 12 h and abstain from exercise for 24 h prior to each scheduled testing session. Subjects reported to the Human Performance Lab for body composition testing. At similar timepoints (i.e., Pre and Post), performance assessments were conducted by the football team’s strength and conditioning personnel.

### 2.3. Offseason Program

The athletes were prescribed a strength and conditioning program designed to promote hypertrophy and improve muscular imbalances. Athletes reported for training four days/week and performed 12–15 exercises per session. Each training session was divided into four phases: movement preparation, primary lifts, auxiliary, and core. The movement preparation phase was performed following a dynamic warm-up and is intended to prepare exercise-specific muscles for the primary lifts. The primary lift phase included power or maximal effort exercises, which could include variations of clean and jerks, vertical jumps, broad jumps, and front and back squats. Auxiliary lifts varied depending on the training focus. The last phase, defined as the core phase, demanded that athletes perform core strengthening and stabilizing exercises.

### 2.4. Body Composition Assessment

Body composition of subjects was first determined by DXA (Hologic Horizon W, Bedford, MA, USA; Apex Software Version 5.6.0.5). The scan was administered with subjects positioned supine in accordance with manufacturer recommendations. Previous between-day test–retest analysis within the laboratory indicated a standard error of measurement (SEM) of 0.43 kg for total body mass (BM), 0.43 kg for total FM, and 0.63 kg for total FFM assessed by DXA [20]. SEM values for DXA segmental FM estimates were 0.23, 0.31, and 0.52 kg for the legs, arms, and trunk, respectively. SEM values for DXA segmental FFM estimates were 0.43, 0.58, and 0.72 kg for the legs, arms, and trunk, respectively. Following analysis by DXA, body composition was assessed by BIA (Biospace InBody 770) in accordance with manufacturer recommendations. Previous between-day test–retest analysis within the laboratory indicated a SEM of 0.54 kg for total BM, 0.49 kg for FM, and 0.64 kg for FFM assessed by BIA. SEM values for BIA segmental FM estimates were 0.16, 0.09, and 0.28 kg for the legs, arms, and trunk, respectively. SEM values for BIA segmental FFM estimates were 0.28, 0.21, and 0.39 kg for the legs, arms, and trunk, respectively.

### 2.5. Performance Assessment

While training volumes differed between position-specific groups (i.e., BIGS and SKILLS), performance testing protocols were similar, consisting of the same exercises: 1-repetition maximum (1-RM) back squat, 3-RM hang clean, 1-RM bench press, 225-pound repetitions to failure bench press test, 1-RM front squat, 1-RM incline bench, 3-RM/1-RM deadlift for BIGS, 3-RM/1-RM trap bar deadlift for SKILLS, 1-RM power clean, broad jump, 40-yard dash, pro agility, and vertical jump. Performance testing was administered at two timepoints, pre and post mesocycle, similar to body composition assessments.

### 2.6. Statistical Analysis

Repeated-measures analysis of variance (ANOVA) tests were performed using the afex R package [21] with time (pre or post) and assessment method (DXA or BIA) specified as within-subjects factors. Normality of residuals was assessed by visual examination of quantile-quantile plots, supplemented by Shapiro–Wilk tests. When normality violations occurred, data were transformed using the BestNormalize R package [22] to achieve a normal distribution. Follow up for significant effects was performed using pairwise comparisons with Tukey adjustment via the emmeans R package [23]. All data in figures and tables are presented as raw (i.e., untransformed) to aid interpretability. Performance changes between the beginning and end of the assessment period were evaluated using paired-samples *t*-tests.

In addition to the analysis of raw data, the change values (i.e., Δ, calculated as the final value minus the baseline value) detected by DXA and BIA were compared. The mean difference (MD) between Δ values was calculated as the BIA value minus the DXA value for each variable. The concordance correlation coefficient (CCC; also known as Lin’s correlation coefficient [24]) and associated 95% confidence interval were calculated using the DescTools R package [25]. The CCC integrates aspects of both precision and accuracy to quantify the deviation of data from the line of identity (i.e., the perfect linear relationship between methods with an intercept of 0 and a slope of 1) [25]. The methods of Bland and Altman [26] were utilized alongside linear regression to assess the degree of proportional bias in Δ values. The 95% limits of agreement (LOA) were also calculated. The total error (i.e., pure error) for Δ values was estimated as:$$\mathrm{TE}=\sqrt{\Sigma {\left(V_{\mathrm{BIA}}-V_{\mathrm{DXA}}\right)}^{2}/n}$$
where V_BIA_ is the BIA estimate for a given variable, and V_DXA_ is the DXA estimate for a given variable. Data were visualized using the ggplot2 R package [27]. Statistical significance was accepted at *p* ≤ 0.05. Data were analyzed using R (v. 4.0.2).

## 3. Results

During the study period, improvements in all performance variables were observed, with the exception of broad jump and pro agility shuttle (Table 1).

Method by time interactions were observed for BM (*p* = 0.03), arms FFM (*p* = 0.03), and legs FFM (*p* = 0.01) (Table 2). Follow up testing indicated that BM increased from pre to post for BIA but not DXA (Figure 1A). Additionally, the BM estimates from BIA and DXA differed at both pre and post. For arms and legs FFM, an increase from pre to post was observed for DXA, but not BIA (Figure 1G,H). Additionally, arms and legs FFM estimates differed between methods at both pre and post. Time main effects indicated an increase in total FFM (Figure 1B) and trunk FFM (Figure 1I) from pre to post. Finally, method main effects indicated higher leg FM values for DXA (Figure 1E) and higher trunk FM values for BIA (Figure 1F). No significant effects were observed for total FM (Figure 1C) or arms FM (Figure 1D). When compared with our laboratory’s between-day SEM values, the observed changes were larger than the SEM for DXA BM, BIA BM, DXA FFM, BIA FFM, DXA FM, DXA leg FFM, BIA trunk FFM. The change in DXA trunk FFM was approximately equivalent to the SEM (0.67 vs. 0.72 kg), as was the change in BIA arms FFM (0.18 vs. 0.21 kg). Changes in BIA FM, BIA leg FFM, DXA arm FFM, and all segmental FM variables estimated by DXA and BIA were smaller than the corresponding SEM values. Individual changes in body composition variables are presented in Figure 2. DXA-derived unadjusted FFMI increased from 23.1 ± 2.1 kg/m^2^ to 23.6 ± 2.1 kg/m^2^ (*p* = 0.0001). When FFMI was further adjusted for height as in previous investigations [3,28,29], the mean increase in FFMI was identical after rounding, although greater variability of values was observed (pre: 23.1 ± 2.4 kg/m^2^; post: 23.6 ± 2.5 kg/m^2^).

Changes in total BM, FFM, and FM were significantly correlated based on CCC values, with the strongest relationship observed for BM (Figure 3). Notable proportional bias was observed for changes in FM, with a smaller magnitude observed for FFM. Changes in arm FFM, but not arm FM, were correlated based on the CCC (Figure 4). Notable proportional bias was also observed for arm FM changes. Changes in leg FFM and FM were correlated, although notable proportional bias was observed for legs FFM (Figure 5). Changes in trunk FFM and FM were not correlated between methods (Figure 6).

## 4. Discussion

The main objective of this study was to compare body composition changes estimated by DXA and BIA in Division III collegiate American football players during an early-offseason training program. As previous studies have shown clear FM and FFM cycling throughout a year-long college football season [1,18], assessment of body composition changes can provide valuable feedback for football practitioners. In the current study, improvements in body composition and performance were observed over a seven-week early-offseason training program. Overall, our hypothesis of no differences between DXA and BIA total body composition changes was partially supported. Both methods detected an increase in FFM without changes in FM. While a statistically significant difference in BM change was detected—with the BIA scale but not DXA detecting a significant increase—the 0.3-kg difference may have been negligible. Whole-body changes in BM, FFM and FM were also correlated, with CCC values of 0.96 for BM and ~0.5 for FFM and FM. However, varying levels of proportional bias were observed.

As broadly hypothesized, some segmental differences between methods were observed, although this was not the case for all variables. Additionally, some segmental body composition variables—particularly FM outcomes—did not change during the intervention. Method by time interactions were observed for arm and leg FFM. Follow up testing indicated that DXA detected a significant increase of 0.39 for arm FFM and 0.56 kg for leg FFM, while BIA did not detect a change in either variable (Δ value of 0.18 and 0.09 kg, respectively). Additionally, DXA estimated larger FFM values at both pre and post timepoints for these variables. Significant method main effects for trunk and leg FM demonstrated a clear difference in segmental attribution of FM between DXA and BIA, with DXA grouping more of the FM in the leg region. While the primary aim of this study was a comparison of body composition assessment methodology, the observed performance changes support the effectiveness of the offseason training program. Improvements in every performance variable except broad jump and pro agility shuttle were observed.

Changes in whole-body BM, FFM, and FM were significantly correlated between DXA and BIA. These results generally support recent findings in resistance training studies by Schoenfeld et al. [16] and Tinsley et al. [17] suggesting that differences between DXA and BIA whole-body estimates may be acceptable in this context. Conversely, the present findings also contrast with previous cross-sectional research evaluating agreement between the two methods in collegiate football players [31]. However, although the previous cross-sectional investigation used the same BIA analyzer (InBody 770), different DXA scanners were employed. It is established that different DXA scanners and software versions influence output [32]. Additionally, while cross-sectional research provides some useful information, some body composition assessment technologies can similarly track changes in body composition longitudinally despite differences in measurements at a single point. When this occurs, longitudinal agreement between technologies relies on consistent over- or underestimation across time. Previously, Raymond et al. [31] observed overestimation of FFM by BIA in comparison to DXA and concluded that BIA is not an accurate method for determining body composition in this population. While the present study observed a similar—but not statistically significant—overestimation of FFM by BIA using the same bioimpedance technology at pre- and post-measurements, both BIA and DXA detected an increase in FFM across time. Additionally, the possible overestimation of FFM by BIA compared to DXA in the present study was likely exacerbated by the concurrent larger BM values detected by the scale component of BIA as compared to DXA. These observations suggest that while body composition estimates from DXA and BIA are not interchangeable, both have utility based on the lack of difference between total FFM changes detected between devices.

Significant increases in FFM were observed in the early offseason period, from January to March. The magnitudes of the observed accretion of FFM during this time period support suggestions in previous research that posits FFM is lost over the course of the competitive season and quickly regained in the early offseason. These results are similar to the findings of Binkley et al. [1], who observed an average ~2.2-kg accretion of lean mass during the spring season (December to May), which followed an average loss of ~1.6 kg of lean mass during the preceding college football season. The present study observed accretion of FFM by DXA and BIA (Δ of 1.57 and 1.15 kg, respectively) over a shorter training program. This time-course of lean mass loss and regain may potentially help explain why Trexler et al. [18], who measured from pre-season through March, did not observe decreases in FFM from pre- to post-season.

Body size and composition are closely related to relevant performance outcomes in American football [33,34]. In previous research comparing FFMI between divisions of college football, Trexler et al. [3] concluded that FFMI was capable of discriminating between Division I and Division II college football players. While this would sensibly extend to Division III due to the competitive advantage FFM may confer in football, the present study finds that the Division III players in this study had similar DXA-derived FFMIs to the Division II players reported by Trexler et al. (23.1 kg/m^2^ at pre and 23.6 kg/m^2^ at post in the present study vs. 23.4 kg/m^2^ in Trexler et al. [3]). However, Division II players measured by Currier et al. [28] had higher FFMIs (24.9 ± 2.4 kg/m^2^) than the Division III players in this study, as well as both the Division I and II players that Trexler et al. measured. Several factors may explain the apparently conflicting evidence for differentiating between divisions using FFMI in collegiate football players. Previous research has determined that offensive and defensive lineman have higher FFMIs than their skill position counterparts [3]. If a sample of football players was disproportionately lineman or other BIGS, such as in the present study, FFMIs of the sample would be biased higher. It is also possible that there is less differentiation in body size, body composition, and performance between players in Division III and Division II than between players in Division II and Division I. However, the similarities between the Division II data from Trexler et al. [3] and the Division III data in the present study may also be due to the caliber of these athletes within their respective division. Specifically, the Division II team that Trexler et al. measured was unranked; in contrast, the team measured in the present study progressed to the Division III NCAA quarterfinals and was ranked as the #1 team in Division III for seven weeks by a reputable sports media company (d3football.com; November 2020). While the current preliminary data suggest that FFMI may not be capable of discriminating between lower competitive divisions, more research is necessary to fully establish potential relationships between FFMI and competitive division, as well as football-specific performance characteristics.

Segmental body composition estimates have important practical applications for football practitioners, as determining regional FFM changes can guide targeted training programs. Since multi-compartment models—widely considered molecular-level criterion methods [35]—are not capable of estimating segmental body composition, DXA and BIA are commonly used when regional information is desired. In the present study, changes in FFM of the arms and legs were only detected by DXA. The extent of the discrepancy of arm and leg FFM accretion in the current study may be practically relevant, as it could lead to different interpretations of whether changes are occurring when viewed by practitioners. Analysis of DXA-derived arm and leg changes in FFM (Δ = 0.39 and 0.56 kg, respectively) during the current study would suggest significant accretions of FFM occurred. In contrast, evaluation by BIA would suggest that no measurable accretion of FFM occurred in the arms and legs (Δ = 0.18 and 0.09 kg respectively). These results question the utility of segmental BIA for detecting appendicular FFM accretion as compared to DXA.

The underlying reason for DXA, but not BIA, detecting appendicular FFM increases could be due to the measurement characteristics of the technologies. DXA, as an imaging technology, may be better able to detect small changes in specific body components since the added mass is visualized as tissue-containing pixels with distinct attenuation ratios within DXA images [36]. In contrast, bioimpedance technologies rely on the bioelectrical properties of body tissues in response to injected current. While it is known that changes in the diameter of body segments and tissue quality influence raw bioimpedance [37], as well as subsequent estimates of body fluids and composition, it is possible that the small changes observed in the present study were insufficient to elicit a discernible change in the bodily response to the applied electrical current. Additionally, consistent segmentation of distinct body segments is readily achievable by DXA through placement of analysis lines based on common boney landmarks. In contrast, segmental BIA relies on proprietary algorithms to determine where transitions between body segments occur, typically without the option for users to review or manually adjust these segments.

The current study observed a clear difference in attribution of FM between the legs and trunk when comparing BIA and DXA. This discrepancy can likely be ascribed to differentiation of the FM in the pelvic region. Segmentation of this region via DXA is demonstrable by examining the regions of interest in DXA analysis software. The standard region of interest lines group a portion of the pelvic area with the legs. In contrast, BIA apparently attributed more FM in this region to the trunk, although the utilization of differential impedance between body segments makes the exact separation points between body segments by BIA less certain than for imaging technologies like DXA. This finding supports earlier analyses in a variety of collegiate athletes [38] and generic populations [39] in which leg FM was similarly underestimated in BIA relative to DXA. Interestingly, in similar investigations in female collegiate athletes [40], this underestimation of leg FM by BIA was not apparent. These findings may indicate that this differential attribution is only demonstrable with a larger segmental mass or that sex-based differences could be present [41]. While this finding could be interpreted to further reduce confidence in BIA for producing accurate segmental body composition estimates in collegiate football players, this is likely a simple technological difference in the segmental attribution of body mass.

There are several limitations to the present study, including the moderate sample size and the utilization of a single BIA analyzer and DXA scanner. Particularly for BIA, notable technological differences are present between specific analyzers, which can affect the quantification of raw bioimpedance, body fluids, and body composition [42,43]. In the present study, we did not attempt to precisely quantify nutritional of dietary supplement intake in study subjects. While the primary purpose of the study was to simply assess the comparability of DXA and BIA for detecting body composition changes in the context of weight gain, nutritional and dietary supplementation information could have helped further contextualize our results. Nonetheless, the observation of clear increases in BM objectively supports that subjects were in an energy surplus during the study [44], in accordance with our goal. Furthermore, the known problems with accuracy of self-reported energy and nutrient intake often preclude high levels of confidence in these data [45]. Additional research using distinct BIA and DXA technologies, as well as larger samples from a variety of sports, should be conducted to further elucidate the comparability of body composition changes estimated by these methods and better inform their use.

Ultimately, according to the findings of this study, BIA may be able to adequately detect increases in whole-body FFM in collegiate American football athletes as compared to DXA. However, BIA may possess less sensitivity in detecting segmental FFM increases, particularly in appendages. Therefore, the purpose of body composition assessment and specific outcome variables being examined should be a consideration when selecting body composition assessment methods for football athletes. For practitioners using segmental BIA to assess FFM accretion in the offseason, caution should be employed when interpreting segmental changes. Additional research is needed to determine if this is also evident in other contexts, such as during in-season FFM loss.

## 5. Conclusions

In collegiate American football players, the early offseason increases in total FFM are detectable by both DXA and BIA. While this indicates the utility of a specific BIA technology for assessing a relevant body composition outcome, shortcomings in the ability to detect segmental changes in FFM were also observed. Specifically, BIA was unable to detect the increases in appendicular FFM demonstrated with DXA. Nonetheless, BIA may present a reasonable alternative to DXA for estimating total FFM changes in American football athletes due to the accompanying cost and time savings, as well as the relative inaccessibility of DXA for many teams. If using body composition changes to help determine programming changes, practitioners are encouraged to understand the limitations of each assessment technology and consider which specific outcome variables should be interpreted for a given technology.

## Figures and Tables

**Figure 1 sports-09-00112-f001:**
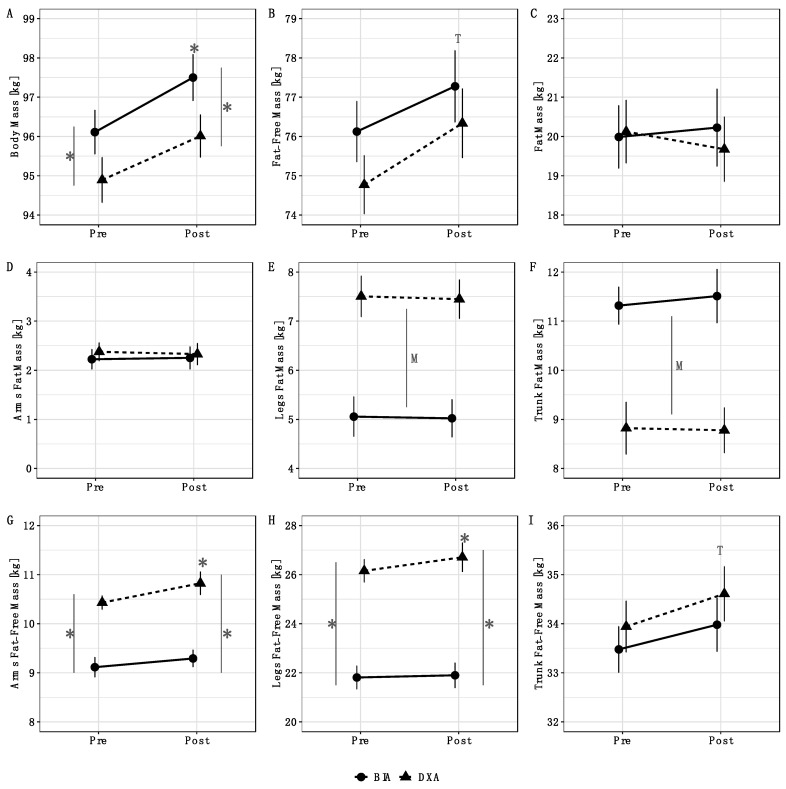
Analysis of variance results. Method by time interactions were observed for BM (*p* = 0.03), arms FFM (*p* = 0.03), and legs FFM (*p* = 0.01). Follow up testing indicated that BM increased from pre to post for BIA but not DXA; additionally, BM estimates from BIA and DXA differed at both pre and post (**A**). For arms and legs FFM, an increase from pre to post was observed for DXA, but not BIA; additionally, arms and legs FFM estimates differed between methods at both pre and post (**G**,**H**). Time main effects indicated an increase in total FFM (**B**) and trunk FFM (**I**) from pre to post. Method main effects indicated higher leg FM values for DXA (**E**) and higher trunk FM values for BIA (**F**). No significant effects were observed for total FM (**C**) or arms FM (**D**). Error bars indicate within-subjects SEs using the Cosineua–Morey–O’Brien method [21,30]. “T” indicates a time main effect, “M” indicates a method main effect, and * indicates a significant difference at a particular time point or a significant change relative to baseline.

**Figure 2 sports-09-00112-f002:**
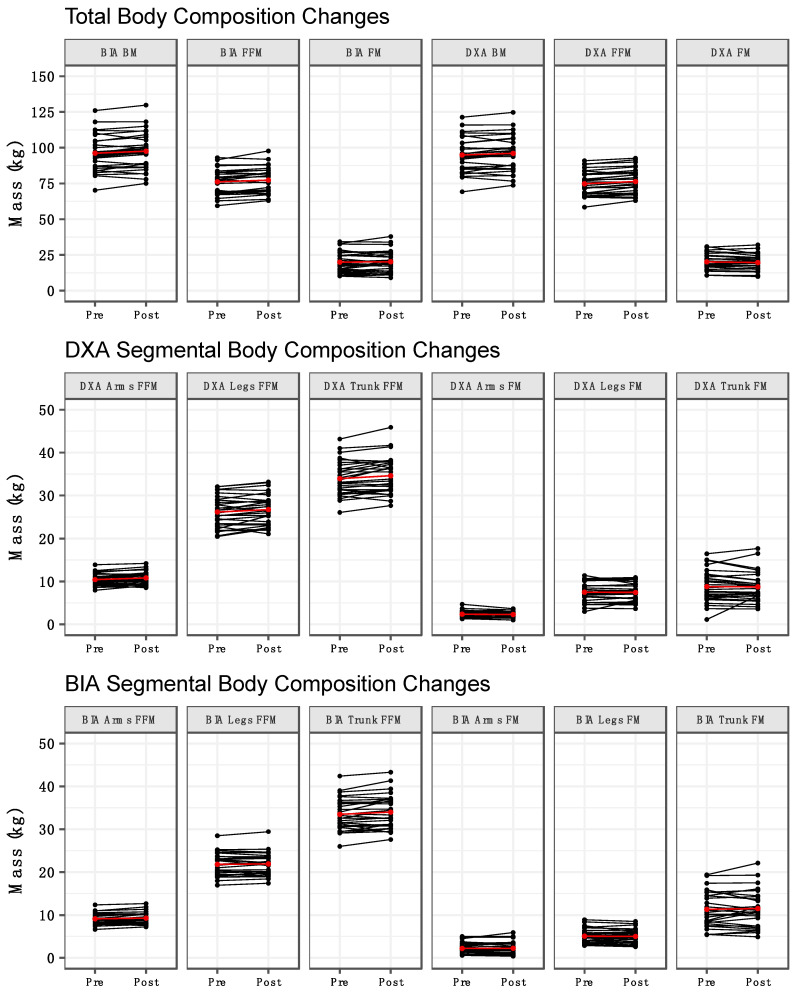
Individual body composition changes. Individual changes in body composition variables are displayed. Black lines represent individual subjects, and the red line represents the mean change.

**Figure 3 sports-09-00112-f003:**
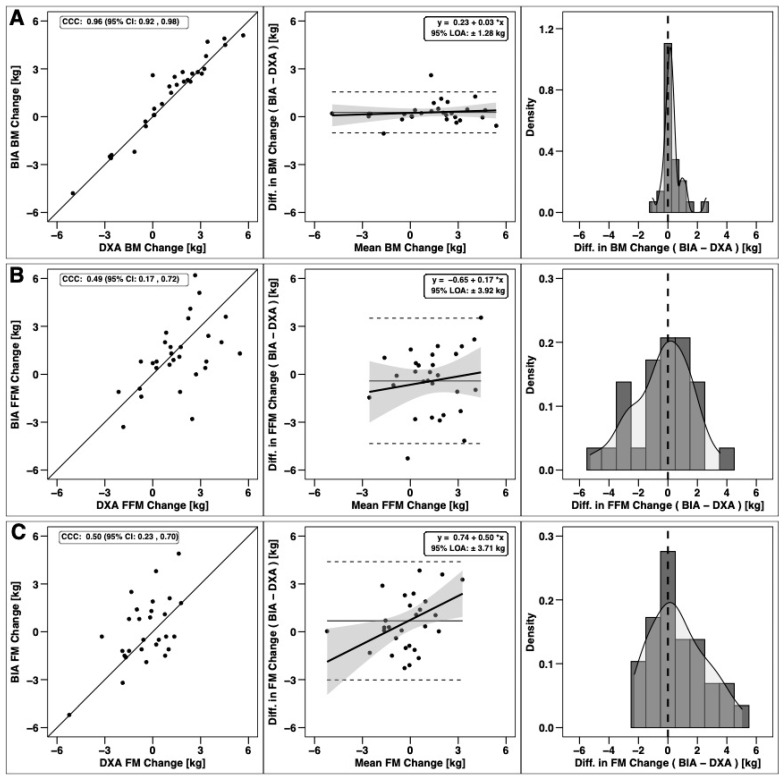
Agreement of total body composition changes. The agreement of changes in body mass (**A**), fat-free mass (**B**), and fat mass (**C**) is displayed. Within each row, the left panel is a scatterplot of individual data points as compared to the line of identity (i.e., the solid black line, which represents perfect agreement); the concordance correlation coefficient (CCC) is also displayed, along with its 95% confidence interval (CI). The middle panel within each row represents results of Bland–Altman analysis paired with linear regression to visualize fixed and proportional bias; the 95% limits of agreement (LOA) and regression equation are displayed. The right panel within each row displays a histogram indicating the occurrence (i.e., density) of varying levels of discrepancies between dual-energy X-ray absorptiometry (DXA) and bioelectrical impedance analysis (BIA) changes.

**Figure 4 sports-09-00112-f004:**
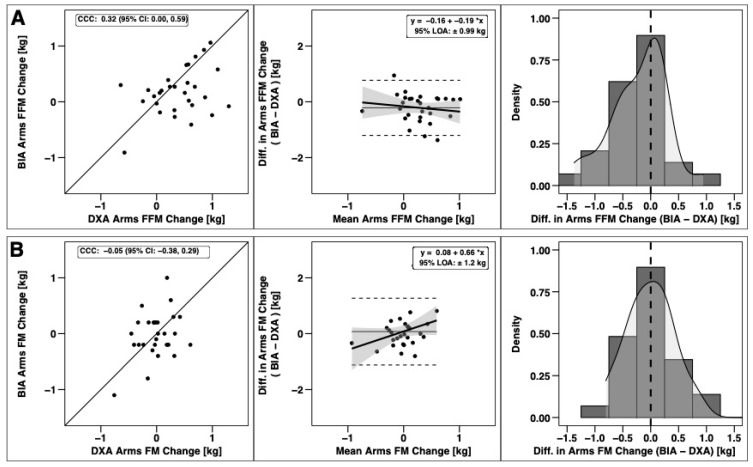
Agreement of Arm Body Composition Changes. The agreement of changes in fat-free mass (**A**) and fat mass (**B**) is displayed. See legend of Figure 3 for explanation of panels.

**Figure 5 sports-09-00112-f005:**
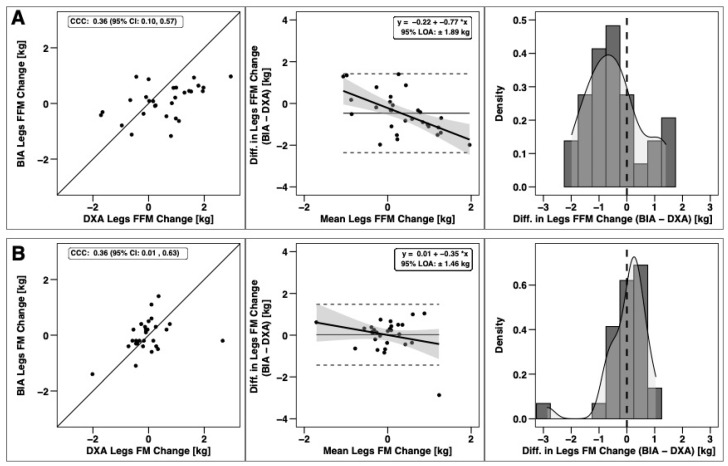
Agreement of leg body composition changes. The agreement of changes in fat-free mass (**A**) and fat mass (**B**) is displayed. See legend of Figure 3 for explanation of panels.

**Figure 6 sports-09-00112-f006:**
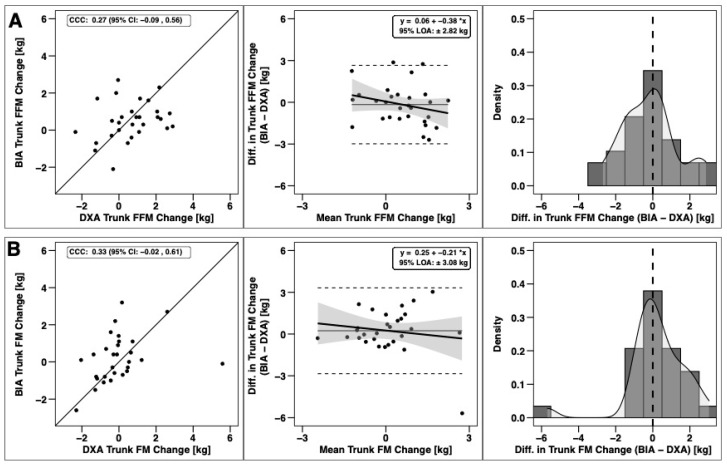
Agreement of trunk body composition changes. The agreement of changes in fat-free mass (**A**) and fat mass (**B**) is displayed. See Legend of Figure 3 for explanation of panels.

**Table 1 sports-09-00112-t001:** Performance Changes.

	*n*	Base	Post	Δ	*p*
Bench Press 1 RM (kg)	19	119.3 ± 18.6	125.5 ± 19.8	6.2 ± 4.5	<0.001 *
Bench Press Reps @ 225	17	6.4 ± 6.3	9.3 ± 7.1	2.9 ± 2.5	<0.001 *
Incline Bench Press 1 RM (kg)	18	100.0 ± 16.2	105.6 ± 14.0	5.6 ± 6.7	0.003 *
Back Squat 1 RM (kg)	17	177.7 ± 34.5	191.4 ± 33.5	13.8 ± 11.9	<0.001 *
Front Squat 1 RM (kg)	18	141.0 ± 22.5	154.4 ± 25.3	13.4 ± 7.4	<0.001 *
Hang Clean 3 RM (kg)	16	99.6 ± 10.9	103.1 ± 10.9	3.6 ± 4.2	0.004 *
40 Yard Dash (s)	19	5.0 ± 0.3	4.8 ± 0.2	−0.2 ± 0.2	<0.001 *
Broad Jump (m)	17	2.6 ± 0.2	2.6 ± 0.2	0.0 ± 0.11	0.42
Vertical Jump (cm)	17	54.6 ± 13.6	63.1 ± 7.3	8.4 ± 12.6	0.01 *
Pro Agility Shuttle (s)	18	4.6 ± 0.3	4.6 ± 0.3	0.0 ± 0.2	0.74

*p* values from paired samples *t*-tests. Statistical significance was accepted at *p* ≤ 0.05, as indicated by *. 1 RM: 1-repetition maximum; 3 RM: 3-repetition maximum.

**Table 2 sports-09-00112-t002:** Agreement of Body Composition Changes.

					ANOVA (Raw Data)		
Variable	DXA Δ (Mean ± SD)	BIA Δ (Mean ± SD)	MD ± SD	TE	*p* (Method)	*p* (Time)	*p* (Method by Time)
BM	1.11 ± 2.43	1.39 ± 2.51	0.27 ± 0.65	0.70	<0.001 *	0.01 *	0.03 *
FM	−0.45 ± 1.54	0.24 ± 2.31	0.69 ± 1.89	1.98	0.92	0.55	0.13
FFM	1.57 ± 1.86	1.15 ± 2.13	−0.41 ± 2.00	2.01	0.07	0.004 *	0.62
Arms FM	−0.05 ± 0.35	0.03 ± 0.48	0.07 ± 0.61	0.60	0.13	0.88	0.99
Legs FM	−0.06 ± 0.73	−0.03 ± 0.57	0.02 ± 0.74	0.73	<0.001 *	0.66	0.88
Trunk FM	−0.04 ± 1.46	0.19 ± 1.27	0.24 ± 1.57	1.56	<0.001 *	0.64	0.69
Arms FFM	0.39 ± 0.47	0.18 ± 0.42	−0.22 ± 0.50	0.54	<0.001 *	<0.001 *	0.03 *
Legs FFM	0.56 ± 1.10	0.09 ± 0.60	−0.47 ± 0.96	1.06	<0.001 *	0.03 *	0.01 *
Trunk FFM	0.67 ± 1.32	0.50 ± 1.04	−0.16 ± 1.44	1.42	0.21	0.02 *	0.74

Δ, MD, and TE values are displayed in kg. *p* values from repeated measures ANOVA. Statistical significance was accepted at *p* ≤ 0.05, as indicated by *. DXA: dual-energy X-ray absorptiometry; BIA: bioelectrical impedance analysis; MD: mean difference; TE: total error; BM: body mass; FM: fat mass; FFM: fat-free mass.

## Data Availability

Data may be available upon reasonable request to the corresponding author, pending relevant approvals.

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
