# Peer review of "Offseason Body Composition Changes Detected by Dual-Energy X-ray Absorptiometry versus Multifrequency Bioelectrical Impedance Analysis in Collegiate American Football Athletes"

_sports, 2021, doi:10.3390/sports9080112_

Round 1
Reviewer 1 Report
The presentation of the data in this study was clear, and the results were in line with the actual situation. However, the insufficient number of people tested limits the value of this research. On the two issues of determining the number of subjects and statistical power, it is recommended that the authors make additional explanations.
The authors mentioned two groups BIGS and SKILLS in the "performance assessment". However, in the manuscript, or in the results, there was no special description of "BIA's characteristics for these two groups"? It is recommended to add additional explanations.
Why was BIA (Inbody 770) less sensitive to “segmental FFM increase”? It is suggested that the authors can add to the discussion.
BIA has many test conditions and restrictions. This is especially true for the standing BIA. It is recommended that authors add "preparation conditions for BIA testing" in the manuscript.
In the results, although the author has clearly stated in the manuscript that the individual body composition of the subjects changes between DXA and BIA. It is recommended that the authors add additional "body composition information of the subject's BIA and DXA before and after the test" in the manuscript.
Participants n=16-19, in the test of exercise ability. For the change in body composition, n=29 was measured. It is recommended that the authors clarify the relationship between the above two different people under test in the manuscript.
Author Response
Thank you to the reviewers for taking the time to review are submitted paper and providing constructive feedback that will improve the overall quality and presentation of the paper.

Reviewer 2 Report
The concept of the manuscript is very interesting and well-written. The statistics were also very detailed and well done. However, there is a major flaw in this paper as diet is totally ignored. Diet plays a part in body composition. Training alone may not attribute to major changes in body composition. In this study, the element of training was used to induce muscle changes. Thereafter, techniques of body composition using machines (BIA vs DXA) was used to measure body composition. If strength training was used to induce muscle gains, why is there no control for diet? If the study was on the accuracy of the machines, the authors could have randomly assessed as many football participants as possible and compare the differences. The major rationale of the study is confusing.
Major Changes in Manuscript
- Line 45-47: "hence the magnified evaluation of the bench press and jump performance at the National Football League (NFL) combine." - sentence not clear, need to explain.
- Line 63: "there is a paucity of research validating BIA against DXA in athletic populations and during periods of purposeful weight gain." - this is an erroneous statement. There is sufficient research between BIA and DXA in a variety of population. Authors need to justify in another manner and not claim paucity of research as a rationale.
- Is there any data on participants' dietary records during the period of testing as it involves body composition? If there is no control for diet, body composition may be affected as football players might consume a lot during the period of testing. So the results might be erroneous.
- Line 125-137: There is so many changes in the body composition values, could it be due to diet, since there is no control in diet?
- Was there a questionnaire asking for supplements consumption (e.g. protein shakes, or ergogenic aids like testosterone or growth hormones, which plays a part in muscle anabolism.)
- Were the participants doing the same workouts as different positions in football require different strength conditioning methods, which may affect the segmented body composition analysis.
Minor Changes in Manuscript
- Include results (mean+/- SD etc) for abstract.
- Place descriptive statistics of the participants under Methods.
- Can explain in further detail for the offseason program (e.g. how long was each duration of training session? How many repetitions did they do?)
- Explain 'BM' again and abbreviate under Body Composition Assessment
- Keep term of 'subjects' or 'participants' consistent throughout manuscript. Do not use both terms.
- Provide exact P value for Table 1.
- What was the significance of the p value for Tables 1 and 2?
- What are the correlation/'r' values for Figures 3 and 4? Refer and cite
Gupta, N., Balasekaran, G., Govindaswamy, V. V., Hwa, C. Y., & Shun, L. M. (2011). Comparison of body composition with bioelectric impedance (BIA) and dual energy X-ray absorptiometry (DEXA) among Singapore Chinese. Journal of Science and Medicine in Sport, 14(1), 33-35. - Label Figure 3 as Figure 3a for top row, Figure 3b for middle row and Figure 3c for bottom row.
- Under Discussion, if the authors mentioned that DXA is a more accurate method of measurement, is it contradictory if it's mentioned "BIA may be adequately detect increases in FFM in collegiate American football athletes as compared to DXA." This study used the BIA analyzer (InBody 770), however, there are other types of BIA analyzers that may use different algorithms to measure body composition that may lead to slightly different results.
- Provide a short conclusion on how the results of offseason body composition changes can affect or help to improve training and performance in American football athletes.
Author Response
Thank you to the reviewers for taking the time to review are submitted paper and providing constructive feedback that will allow us to improve the overall quality and presentation of our findings.

Round 2
Reviewer 2 Report
NA